# Elevated Circulating and Placental SPINT2 Is Associated with Placental Dysfunction

**DOI:** 10.3390/ijms22147467

**Published:** 2021-07-12

**Authors:** Ciara N. Murphy, Susan P. Walker, Teresa M. MacDonald, Emerson Keenan, Natalie J. Hannan, Mary E. Wlodek, Jenny Myers, Jessica F. Briffa, Tania Romano, Alexandra Roddy Mitchell, Carole-Anne Whigham, Ping Cannon, Tuong-Vi Nguyen, Manju Kandel, Natasha Pritchard, Stephen Tong, Tu’uhevaha J. Kaitu’u-Lino

**Affiliations:** 1The Department of Obstetrics and Gynaecology, Mercy Hospital for Women, The University of Melbourne, Heidelberg, VIC 3084, Australia; spwalker@unimelb.edu.au (S.P.W.); teresa.mary.macdonald@gmail.com (T.M.M.); emerson.keenan@unimelb.edu.au (E.K.); nhannan@unimelb.edu.au (N.J.H.); m.wlodek@unimelb.edu.au (M.E.W.); aroddymitchell@gmail.com (A.R.M.); drcwhigham@gmail.com (C.-A.W.); ping.cannon@unimelb.edu.au (P.C.); tuong-vi.nguyen@unimelb.edu.au (T.-V.N.); manju.kandel@unimelb.edu.au (M.K.); natashalpritchard@gmail.com (N.P.); stong@unimelb.edu.au (S.T.); t.klino@unimelb.edu.au (T.J.K.-L.); 2Mercy Perinatal, Mercy Hospital for Women, Heidelberg, VIC 3084, Australia; 3The Department of Anatomy and Physiology, The University of Melbourne, VIC 3010, Australia; jessica.briffa@unimelb.edu.au; 4Manchester Academic Health Science Centre, St Mary’s Hospital, University of Manchester, Manchester M13 OJH, UK; Jenny.Myers@manchester.ac.uk; 5The Department of Physiology, Anatomy and Microbiology, La Trobe University, Bundoora, VIC 3086, Australia; t.romano@latrobe.edu.au

**Keywords:** placental insufficiency, SPINT2/HAI-2, preeclampsia, foetal growth restriction, intrauterine growth restriction, small for gestational age

## Abstract

Biomarkers for placental dysfunction are currently lacking. We recently identified SPINT1 as a novel biomarker; SPINT2 is a functionally related placental protease inhibitor. This study aimed to characterise SPINT2 expression in placental insufficiency. Circulating SPINT2 was assessed in three prospective cohorts, collected at the following: (1) term delivery (*n* = 227), (2) 36 weeks (*n* = 364), and (3) 24–34 weeks’ (*n* = 294) gestation. SPINT2 was also measured in the plasma and placentas of women with established placental disease at preterm (<34 weeks) delivery. Using first-trimester human trophoblast stem cells, SPINT2 expression was assessed in hypoxia/normoxia (1% vs. 8% O2), and following inflammatory cytokine treatment (TNFα, IL-6). Placental SPINT2 mRNA was measured in a rat model of late-gestational foetal growth restriction. At 36 weeks, circulating SPINT2 was elevated in patients who later developed preeclampsia (*p* = 0.028; median = 2233 pg/mL vs. controls, median = 1644 pg/mL), or delivered a small-for-gestational-age infant (*p* = 0.002; median = 2109 pg/mL vs. controls, median = 1614 pg/mL). SPINT2 was elevated in the placentas of patients who required delivery for preterm preeclampsia (*p* = 0.025). Though inflammatory cytokines had no effect, hypoxia increased SPINT2 in cytotrophoblast stem cells, and its expression was elevated in the placental labyrinth of growth-restricted rats. These findings suggest elevated SPINT2 is associated with placental insufficiency.

## 1. Introduction

Aberrations in placentation, particularly those amounting to restricted vascular remodelling, are associated with a constellation of obstetric complications, with significant implications for mothers and babies. This includes foetal growth restriction (FGR), in which affected foetuses fail to achieve their unique growth potential in utero, owing to inadequate uteroplacental perfusion. This confers an increased risk of perinatal morbidity and mortality upon the foetus [1]; in fact, FGR is recognised as the single largest risk factor for stillbirth [2]. Alternatively or simultaneously, placental insufficiency may manifest maternally as preeclampsia, which is characterised by persistent maternal hypertension and end organ dysfunction, arising from vascular endothelial injury [3].

Currently, there is an absence of effective and targeted treatments for both preeclampsia and FGR, and the latter, in particular, eludes precise diagnosis. Consequently, there are no interventions to rescue a poorly functioning placenta; except for iatrogenic preterm delivery, which has its own associated risks. 

In the search for circulating biomarkers of placental insufficiency, SPINT1, a serine protease inhibitor, also called HGF activator inhibitor 1 (HAI-1), has been identified as a promising candidate [4]. SPINT1 is highly expressed in the placenta, where it is localised to the cell surface of villous cytotrophoblasts, and secreted in a proteolytically truncated form [5] through ectodomain shedding [6] into the maternal circulation. By virtue of its dual Kunitz domains, SPINT1 inhibits the activity of proteolytic substrates that are critical for normal placentation. Therefore, SPINT1 mediates the trophoblast secretion of degradative enzymes (serine proteinases, metalloproteinases, and collagenases), which regulate the invasion and remodelling of endometrial spiral arteries [7,8]. Inadequate or superficial remodelling may result in a suboptimal placenta, through oxidative stress-induced placental growth suppression [9], and intermittent placental perfusion, which leads to ischaemia-reperfusion injury [10]. We have recently demonstrated that SPINT1 is reduced in FGR placentas and is modulated by hypoxia; both in human placental cells and in a mouse model of FGR [4].

The validation of this serine peptidase inhibitor as an indicator of placental dysfunction justifies the investigation of its analogue SPINT2/HAI-2. To date, SPINT2 has not been thoroughly assessed in the human placenta; thus, its application in FGR diagnosis is yet to be explored. SPINT2 has a comparable tissue distribution to SPINT1 [11], and contains two extracellular inhibitory Kunitz domains, making it structurally similar to SPINT1. As with SPINT1, SPINT2 regulates matriptase [12], a transmembrane serine protease that is responsible for the degradation of the extracellular matrix components fibronectin and laminin [13]. The involvement of SPINT2 with matriptase in the placenta suggests its importance in placental development [12,14]. Indeed, in mouse placentas, SPINT2 is expressed for the duration of development in the placental labyrinth layer (the site of murine foetomaternal exchange) [12,15]. As reported in *Spint1* knockout mouse models, *Spint2*-deficient mice suffer from placental defects. However, the effects of the loss of *Spint2* extend beyond the placenta, causing embryonic lethality, unless matriptase is simultaneously ablated, in which case there are impairments to the neural tube closure [12].

Given the previous findings of dysregulated SPINT1 in FGR, it was hypothesised that the expression of SPINT2 would be similarly deranged in the placenta and maternal circulation of pregnancies that are complicated by FGR, even prior to diagnosis. Further, the expression of SPINT2 in placental trophoblasts was expected to be regulated by hypoxia and inflammation, which are signature contributors underlying placental insufficiency. Therefore, the aim of this study was to characterise the expression of SPINT2 at the mRNA and protein level in placental and plasma samples from pregnancies that have been affected by preeclampsia and/or FGR, as well as in a rodent model of placental insufficiency, and to observe any hypoxic- or inflammation-mediated changes in expression in vitro.

## 2. Results

### 2.1. SPINT2 Expression Is Deranged in Placental Dysfunction

Given that SPINT2 has not previously been analysed in the human placenta, this study first characterised its expression in pregnancies that were known to be compromised by FGR and/or preeclampsia.

In preterm FGR placentas (*n* = 14), SPINT2 mRNA expression (Figure 1a) was highly variable and did not significantly differ from the controls (*n* = 19). In contrast, the placentas from pregnancies that were complicated by both preeclampsia and FGR (*n* = 20) had significantly decreased SPINT2 mRNA expression (68% of control, *p* = 0.002), whereas those affected by preeclampsia only (*n* = 60) had significantly increased SPINT2 mRNA levels (119% of control, *p* = 0.03).

The level of SPINT2 protein expression was also measured in these placentas (Figure 1b), which revealed no change in FGR placentas from normotensive pregnancies, relative to the controls. In placentas affected concurrently with preeclampsia and FGR (median = 195.4 pg/mL, IQR: 157.0–232.8 pg/mL), SPINT2 was significantly elevated (*p* = 0.0171) compared to the controls (median = 157.6 pg/mL, IQR: 118.4–183.5 pg/mL), with a similar result observed in the placentas from pregnancies that were affected by preeclampsia alone (*p* = 0.0042, median = 189.9pg/mL, IQR: 165.6–226.3 pg/mL).

### 2.2. Circulating SPINT2 in FGR and/or Preeclampsia

We next sought to measure SPINT2 within the maternal circulation in prospective cohorts, prior to any diagnoses.

In maternal plasma (collected upon presentation to MHW for caesarean section at term; Figure 2a), SPINT2 was modestly elevated (*p* = 0.0507) in those women whose infant was born small for the gestational age (SGA, birthweight < 10th centile; *n* = 75, median = 4020 pg/mL), compared to appropriate for the gestational age (AGA, birthweight > 10th centile) controls (*n* = 152, median = 3407 pg/mL). Circulating SPINT2 was significantly elevated (*p* = 0.002) in the SGA (*n* = 128, median 2109 pg/mL, IQR 1355–3069 pg/mL) samples that were collected at 36 weeks’ gestation (Figure 2b), compared to the AGA controls (*n* = 182, median = 1614 pg/mL, IQR: 1139–2360 pg/mL). This association was lost, however, earlier in gestation, where 24- to 34-week plasma from women with underlying vascular disease (MAViS clinic samples, Figure 2c, Appendix A) demonstrated no difference between the AGA control (*n* = 179) and SGA (*n* = 58) levels of SPINT2. In this cohort, SPINT2 did not vary across gestation (Figure 2c) in the controls, but there was a trend towards a modest increase (*p* = 0.054) in SPINT2 across gestation in those women who were destined to birth an SGA infant (R^2^ = 0.0645).

At 36 weeks’ gestation (Figure 2d), circulating SPINT2 was also elevated (*p* = 0.03) in women who were destined to develop term preeclampsia (*n* = 23, median = 2233 pg/mL, IQR: 1643–2661 pg/mL), relative to the controls (*n* = 182, median 1644 pg/mL, IQR: 1218–2480 pg/mL). In the 24- to 34-week plasma from women attending the MAViS clinic (Figure 2e, Appendix A), there was no difference in SPINT2 levels in those who were ultimately diagnosed with preeclampsia, relative to the controls, nor did the protein concentration change, relative to gestation, regardless of the disease status (Figure 2e). 

Interestingly, there were no significant differences in circulating SPINT2 in patients delivering preterm for preeclampsia or FGR, relative to the controls (Figure 2f). 

### 2.3. Hypoxic Regulation of SPINT2 in Trophoblasts

Placental insufficiency is often associated with intermittent placental hypoxia; thus, we assessed the effect of hypoxia on SPINT2 expression.

In primary trophoblasts that were isolated from term placentas, SPINT2 mRNA transcripts were significantly increased in response to hypoxia (Figure 3a; mean = 193% of control, *p* = 0.002), while secreted SPINT2 was unchanged (Figure 3b). In contrast, hypoxia had no effect on SPINT2 mRNA expression in first-trimester cytotrophoblast stem cells (Figure 3c), but SPINT2 protein secretion (Figure 3d) was significantly increased (mean = 412.8% of control, *p* = 0.008). Given SPINT2 is also likely expressed in syncytiotrophoblast, we measured expression and secretion in syncytialised first-trimester human trophoblast stem cells (hTSCs; Figure 3e), observing that oxygen tension had no effect on mRNA expression, but modestly decreased SPINT2 secretion (Figure 3f; mean = 85.6% of control, *p* = 0.008) in the syncytiotrophoblasts that were exposed to hypoxia.

The derangement of SPINT2 expression in response to hypoxia was also demonstrated in placentas from a rat model of uteroplacental insufficiency, induced by ligating the uterine vessels, thereby impeding placental perfusion. There are distinct morphological differences (see Furukawa et al., 2011 [15]) between the rat and human placenta, thus we separated the basalis and labyrinthine layers for analysis of SPINT2. In the basalis region of the restricted placentas (Figure 3g), SPINT2 mRNA expression was significantly depressed (median = 76.9% of control, *p* = 0.004), compared to that of dams who underwent sham surgery. Interestingly, the labyrinth layer—akin to the chorionic villi (including syncytiotrophoblasts, villous cytotrophoblast, stroma and blood vessels) of the human placenta—of restricted placentas had modestly upregulated SPINT2 mRNA expression (median = 108.4% of control, *p* = 0.04).

### 2.4. SPINT2 Is Not Regulated by Inflammation

Preeclampsia is associated with placental and systemic inflammation, and we therefore assessed whether SPINT2 is influenced by pro-inflammatory stimuli. In first-trimester cytotrophoblasts (Figure 4a,c) and syncytialised trophoblasts (Figure 4e,g), we observed no significant effect on SPINT2 mRNA expression. SPINT2 secretion was also unchanged in both the cell types (Figure 4b,d,f), with only low doses of IL-6 stimulating a modest decrease (*p* < 0.01) in syncytiotrophoblasts (Figure 4g).

## 3. Discussion

Throughout the first trimester of pregnancy, the intricate foetal–maternal interface is established through the process of placentation, which involves tightly regulated and complex pathways, the understanding of which is at present incomplete. The accumulation of anomalies in this process can lead to a dysfunctional placenta, which inadequately supplies the foetus, and can have serious consequences for the mother and baby, including preeclampsia and/or FGR. In this study, we sought to characterise the expression of SPINT2 in the placentas and maternal circulation of pregnancies that were complicated by preeclampsia and/or FGR, using three prospective cohorts, a rodent model, in trophoblasts isolated from human tissue at term, and in stem cells from the first trimester. 

By measuring circulating SPINT2 levels in the cases of established disease (delivered at <34 weeks’ gestation), it was apparent that SPINT2 expression is not dysregulated in FGR-like functionally related homologue SPINT1 [4]. While derangements in SPINT2 expression were apparent in the cases of placental insufficiency-mediated pregnancy complications, the measured fluctuations do not reliably reflect the disease status, making SPINT2 an overall poor biomarker candidate. Indeed, an association between circulating SPINT2 and placental insufficiency is apparent only at term (and near term, from 36 weeks onwards), with no distinction between the cases and controls in weeks 24–34, nor at preterm delivery (<34 weeks). As such, SPINT2 lacks the robust predictive potential of its relative, SPINT1 [4]. This is perhaps unsurprising, because, despite their similarities, SPINT2 is more ubiquitously expressed than SPINT1, and their encoding genes are located on different chromosomes (15 and 19, respectively). So, although they likely share a common ancestor gene, they have evolved distinctly [16]. Nevertheless, there is likely involvement of SPINT2 in placental function—given the changes in placental SPINT2 expression that have been observed. 

The expression of SPINT2 mRNA in the preterm placentas was decreased in concurrent FGR and preeclampsia, but elevated where only preeclampsia is present (i.e., with AGA); whereas, SPINT2 protein expression was increased in all cases of preeclampsia (with and without FGR). It is interesting that SPINT2 mRNA and protein are both significantly elevated in preeclamptic (without FGR) placentas, whereas the placentas plagued by concurrent preeclampsia and FGR had decreased SPINT2 mRNA, but elevated protein. This is an unusual phenomenon, and the reason for this disparity has not been elucidated by the present study. It may suggest that there are post-transcriptional modifications in the preeclamptic/FGR placentas, to reduce SPINT2 protein turnover, as a compensatory mechanism for the low transcript levels; however, further study is needed to confirm this hypothesis.

In the cases of FGR, the circulating levels of SPINT2 were modestly elevated at term and at 36 weeks’ gestation; however, at earlier gestations, there were no observed changes in expression. Similarly, there was elevated circulating SPINT2 at 36 weeks in those women who were destined to develop preeclampsia, while earlier gestation levels were unchanged, relative to the controls. Interestingly, SPINT2 was not altered in preterm disease, raising the possibility that derangements in circulating levels only arise in later-onset disease. Alternatively, the lack of statistical significance in the established disease plasma cohort may be attributable, in part, to having relatively few FGR samples (especially plasma; *n* = 6), and the high variability in expression for the samples that were available. A larger sample size would aid in verifying whether there are bona fide alterations in mRNA, and/or protein in the placentas and plasma of FGR-affected pregnancies. 

Frequently, the dysfunctional placenta suffers from suboptimal perfusion and high levels of inflammatory factors, and, in previous research, SPINT2 has been shown to be regulated by hypoxia (in breast cancer cells) [17]. Having found SPINT2 levels to be elevated in placental insufficiency, we used in vitro methods to examine the effect of hypoxia and pro-inflammatory cytokines on the trophoblasts of the placenta, early in gestation and at term. SPINT2 expression did appear to be regulated by hypoxia in first-trimester human trophoblast stem cells (hTSCs), term primary trophoblasts, and in a rat model of restricted placental perfusion. Early in gestation (as modelled by hTSCs), SPINT2 secretion was elevated under hypoxia in cytotrophoblast stem cells, while being decreased in syncytiotrophoblasts. However, there were no changes in transcription at this early stage, with SPINT2 mRNA largely unchanged by oxygen tension. In contrast, in primary trophoblasts, isolated from placentas that were delivered at term, there was an increase in SPINT2 mRNA expression, but no change in protein secretion. This indicates that in response to hypoxic conditions early in gestation, trophoblasts alter the secretion of SPINT2; whereas, nearer to term, transcriptional changes dominate the response to hypoxia. The mechanism behind this difference is uncertain and requires further investigation. Importantly, fluctuations in SPINT2 expression, in relation to hypoxia, were also measured in rat placentas with induced uteroplacental insufficiency, with inverse changes in the basalis and labyrinth zones of the placenta. Notably, SPINT2 mRNA was upregulated in the labyrinthine region of restricted placentas, complementing the findings in term trophoblasts exposed to hypoxia. 

Mouse models have previously established the importance of SPINT2 in placental development [15], and, consequently, embryonic survival. The findings presented here suggest that SPINT2 has a similar importance in human placentation, owing to its derangement in FGR and preeclamptic pregnancies. 

## 4. Materials and Methods

### 4.1. Tissue and Blood Collection at Time of Preterm Delivery from Women with Established Placental Disease (Day of Delivery at <34 Weeks)

To characterise the expression of SPINT2 in the maternal circulation and placenta of FGR- and/or preeclampsia-complicated pregnancies, human specimens were obtained. All studies were approved by the Mercy Health Human Research ethics committee (R11/34).

Placental tissue samples were collected from consenting women, delivering by caesarean section at less than 34 weeks’ gestation, with decision for delivery made independently by the treating obstetric team. The samples were classified as FGR (*n* = 14), PE (*n* = 60) or both (*n* = 20), according to preeclampsia guidelines from ACOG 2020 [18] and FGR defined as birthweight <10th centile on local birthweight charts [19]. The control samples (*n* = 19) were gestation-matched and obtained from women who were delivered preterm due to other complications not associated with placental insufficiency or hypertensive disorders of pregnancy, such as placenta praevia or spontaneous preterm rupture of membranes. Although the control sample comprises pregnancies with complications, gestation-matching is important when investigating proteins highly expressed in the placenta, as the pattern of expression commonly varies with advancing gestation. Samples in both the case and control group were excluded if there were congenital anomalies and/or histopathological evidence of congenital infection. Patient characteristics are detailed in Appendix A.

Tissue was collected and processed within 30 min of delivery by caesarean section. Segments of tissue were dissected and washed in PBS, then samples of roughly equal size were immersed in RNAlater TM stabilisation solution (Thermo Fisher Scientific; Waltham, MA, USA) for 48 h, then snap frozen and stored at −80 °C. Subsequently, RNA or protein was extracted from tissue lysates.

Plasma samples were also collected on the day of delivery at less than 34 weeks’ gestation from women delivering prematurely with FGR (*n* = 6), PE (*n* = 40) or both (*n* = 11). These were compared to gestation-matched blood specimens collected from control pregnancies delivered at term (*n* = 26). These samples were aliquoted and stored at −80 °C until future analysis. Patient characteristics of this cohort are detailed in Appendix A.

### 4.2. Prospective Case-Cohorts

#### 4.2.1. Day of Delivery at Term—FLAG2

The Fetal Longitudinal Assessment of Growth 2 (FLAG2) study recruited 562 unselected women on the day of elective caesarean section at the Mercy Hospital for Women (MHW, Melbourne, Australia). Women who were aged over 18 years with a well-dated singleton pregnancy, at 36^+0^–42^+0^ weeks’ gestation, were eligible to participate. Exclusion criteria included any suspicion of major foetal anomaly or infection; ruptured membranes; labouring women; those who had undergone cervical ripening or steroid administration before the caesarean section; and those who were positive for hepatitis B, C or HIV. A study blood sample was taken at the time of intravenous cannula placement and birthweight centile was determined using the GROW Bulk centile calculator (v8.0.4, 2019). The FLAG2 study was approved by the Mercy Health Research ethics committee (ethics approval number R11/34) and written informed consent was obtained from all participants. The total number of remaining samples used for SPINT2 analysis was 227, comprising 152 controls (appropriate for gestational age, AGA) and 75 cases (SGA). Patient characteristics are shown in Appendix A.

#### 4.2.2. BUMPS—36 Weeks’ Gestation

The Biomarker and Ultrasound Measures for Preventable Stillbirth (BUMPS) study is a large prospective cohort collection at MHW, with samples collected from an unselected population at 28 and 36 weeks’ gestation. Women were screened for eligibility and invited to participate at their oral glucose test, universally offered to non-diabetic pregnant women around 28 weeks’ gestation to test for gestational diabetes mellitus. Following written informed consent, women aged over 18 years, with a singleton pregnancy and normal mid-trimester foetal morphology examination were eligible to participate. The BUMPS study was approved by the Mercy Health Research ethics committee (ethics approval number 2019-012). For this study, a case-cohort of 364 samples was selected from the first 1000 BUMPS participants, including all cases delivering an infant <10th centile (SGA; *n* = 198) according to the GROW Bulk centile calculator (v8.0.4, 2019), all cases delivering with preeclampsia (defined according to ACOG guidelines; *n* = 23) and a cohort of controls (*n* = 182). Patient characteristics detailed in Appendix A.

#### 4.2.3. MAViS—24–34 Weeks’ Gestation

SPINT2 was also measured in a high-risk cohort of patients at the Manchester Antenatal Vascular Service (the MAViS clinic; Manchester, UK). Women are referred to the clinic in early pregnancy for monitoring across gestation based on hypertensive disease, which predisposes to preeclampsia and/or FGR, allowing for longitudinal sampling between 24- and 34-weeks’ gestation. The inclusion criteria for women in the MAViS study were as follows: 1. chronic hypertension (BP ≥ 140/90 at ≤20 weeks; 2. chronic hypertension requiring antihypertensive treatment from ≤ 20 weeks; 3. pre-gestational diabetes with evidence of vascular complications (hypertension, nephropathy); 4. history of ischaemic heart disease; and 5. previous early onset preeclampsia. A case-cohort of 294 participants was recruited between October 2011 and December 2016, with a plasma sample obtained between 24 and 34 weeks, and complete outcome data were included in the current study. These participants were selected from an overall cohort of 518 participants and included 179 control women and 115 who either delivered with preeclampsia, FGR or both. The study was granted ethics approval by the NRES Committee North West (11/NW/0426). Patient characteristics are listed in Appendix A.

### 4.3. Placental Samples from a Rat Model of Placental Insufficiency

In order to assess the in vivo expression of SPINT2, samples were obtained from a previously established rodent model of FGR. The placental deficiency in this model was induced during late gestation (at day 18 of 22), providing an in vivo model of late-onset, placental-derived FGR [20]. Uteroplacental insufficiency was induced by means of bilateral uterine vessel ligation (of both the artery and vein), to restrict the blood and nutrient supply to the foetuses. The control group underwent sham surgery mimicking this procedure, without the ligation of uterine vessels. The details of this protocol can be found in Wlodek et al. (2005) [21].

The placentas were collected, weighed and the labyrinthine layer separated from the basalis layer before being immediately frozen in liquid nitrogen, then stored at −80 °C for later analysis. RNA was extracted from both regions. The development of this model of uteroplacental insufficiency in rats was approved by the La Trobe animal ethics committee (AEC: 12–42), in accordance with the National Health and Medical Research Council’s (NHMRC) Australian code for the care and use of animals for scientific purposes.

### 4.4. Human Trophoblast Stem Cells (hTSCs)

To examine SPINT2 in response to hypoxia and inflammatory stressors, first-trimester human trophoblast stem cells (hTSCs) were obtained from the RIKEN BRC through the National BioResource Project of the MEXT/AMED (Japan), as previously detailed in the manuscript from Okae et al., 2018. This cell line was isolated from first-trimester placentas under ethical approval from Tohoku University School of Medicine [22]. The cells were then cultured in specialised media, according to the optimised conditions in Okae et al., 2018 [22]. Given the localisation of SPINT2 to both cytotrophoblast and syncytiotrophoblast in the placenta [23], some cells were propagated as multipotent cytotrophoblasts, while others were directed to differentiate into the syncytiotrophoblast lineage.

### 4.5. Term Primary Cytotrophoblast Isolation

Primary cytotrophoblast cells were isolated from term placentas according to the protocol optimised by Kaitu’u-Lino et al., 2014 [24]. In summary, a segment of placenta was resected, washed, mechanically dissociated, and enzymatically digested, allowing for the collection of the isolated cells in the supernatant [24]. Cells were then cultured in preparation for subsequent analysis.

### 4.6. Simulation of Trophoblast Hypoxia

Hypoxic conditions were simulated for first-trimester hTSCs and term trophoblasts to assess the effect of inadequate oxygen perfusion during placentation and approaching term, respectively. After a 24-h incubation in 8% oxygen at 37 °C, allowing cells to adhere to the basement membrane (iMatrix-511 for hTSCs, fibronectin for primary trophoblasts), cells were incubated in different oxygen concentrations. Given the physiologically relevant oxygen tension in utero is 8%, those cells designated normoxic were incubated at 37 °C in 8% oxygen for 48 h. Hypoxia involved exposure to 1% oxygen for the same duration. The media was collected for subsequent analysis and cells lysed for RNA extraction.

### 4.7. Simulation of Placental Inflammation

Two inflammatory cytokines, tumour necrosis factor alpha (TNFα; Life Technologies, Carlsbad, CA, USA) and interleukin-6 (IL-6; In Vitro Technologies, Noble Park, VIC, Australia), were added to the media of primary trophoblasts and first-trimester hTSCs (cytotrophoblasts and syncytialised stem cells) to simulate the inflammation common to the preeclamptic placenta. These cells were incubated at 37 °C for 24 h after plating, followed by treatment with 0 ng/mL, 0.1 ng/mL, 1 ng/mL or 10 ng/mL of the recombinant cytokine, diluted with fresh media. After being cultured for a further 24 h in the treatment media, cells and media were collected.

### 4.8. Protein Extraction

Protein was isolated from placental tissue and syncytiotrophoblast, using RIPA buffer containing protease inhibitor cocktail (Sigma-Aldrich; St. Louis, MO, USA) and Halt™ phosphatase inhibitor cocktail (Thermo Fisher Scientific; Waltham, MA, USA) to lyse cells and centrifugation to pellet debris. To quantify the protein content of each sample, a Pierce™ BCA assay (Thermo Fisher Scientific) was performed according to the manufacturer’s protocol. Equal protein amounts were loaded for ELISA.

### 4.9. RNA Extraction

The Genelute™ mammalian total RNA miniprep kit (Sigma-Aldrich; St Louis, MO, USA) was used to extract RNA from cultured hTSCs, primary term trophoblasts and placental tissue, as per the manufacturer’s protocol.

### 4.10. Reverse Transcription

RNA extracted from samples was converted to cDNA using the Applied Biosystems™ high-capacity cDNA reverse transcription kit (Thermo Fisher Scientific; Waltham, MA, USA), following the manufacturer’s guidelines. The reaction comprised 150 ng RNA solution (appropriately diluted with DEPC-treated H_2_O). The iCycler iQ™5 (Bio-Rad, Hercules, CA, USA) protocol was run according to kit specifications, being held at 4 °C after completion until collection and storage at −20 °C for later PCR analysis.

### 4.11. Real-Time Polymerase Chain Reaction (RT-qPCR)

Quantitative PCR was carried out to ascertain the mRNA expression of *Spint1* and *Spint2*, relative to reference housekeeper genes. TaqMan gene expression primers (Thermo Fisher Scientific; Waltham, MA, USA) specific to the genes of interest are detailed in Appendix A, including the appropriate housekeeping genes for each sample set. All PCRs were performed on the CFX384 (Bio-Rad). The average C_t_ of sample duplicates were normalised to appropriate reference genes before being calibrated to the average C_t_ of experimental controls, allowing the results to be expressed as percentage relative to controls.

### 4.12. Enzyme Linked Immunosorbent Assays (ELISAs)

SPINT2 protein levels were measured in maternal plasma samples, hTSC media, and placental lysates via ELISA. The large cohort analyses were analysed using an ELISA kit for SPINT2 (Sigma-Aldrich, St. Louis, MO, USA), following manufacturer’s specifications. The <34 week plasma was diluted 1:12 for SPINT2. 

Cellular SPINT2 in placental lysates was analysed using a SPINT2 DuoSet^®^ ELISA (R&D Systems; Minneapolis, MN, USA). Then, 5 μg of each sample was loaded, diluted in 1% BSA in PBS according to the concentration determined by the BCA assay. Cultured hTSC media was also analysed using the R&D Systems SPINT2 kit. Media samples were undiluted, with the exception of the hypoxia studies, which were diluted 1:2 with 1% BSA in PBS. 

### 4.13. Statistical Analysis

In vitro experiments were carried out in technical triplicate and repeated 3–5 times. The results of in vitro experiments were normalised to controls so data could be expressed as % control. Using GraphPad Prism 8 (GraphPad Software, Inc., San Diego, CA, USA), statistical analyses were carried out, with data first assessed for Gaussian distribution, then analysed using appropriate statistical tests. Maternal characteristics and birth outcome data (Appendix A were compared for all women who were preeclamptic and/or delivered an SGA baby against controls using Mann–Whitney U, unpaired t, Fisher’s exact or Chi-square tests. For all other data, when two groups were compared, a Student’s *t*-test or Mann–Whitney U test was used according to Gaussian distribution. For more than two groups, a one-way ANOVA or Kruskal–Wallis test was used, according to Gaussian distribution, and post hoc analyses ascertained by Dunn’s multiple comparisons test. Outliers were identified and accounted for using a ROUT test.

## 5. Conclusions

Unlike SPINT1, circulating SPINT2 is not consistently dysregulated in diseases of placental insufficiency—in preeclampsia or foetal growth restriction. We have shown that SPINT2 is unlikely to be a clinically useful biomarker; however, we did identify changes in placental SPINT2, which suggest it may be functionally involved in human placentation; this is a role yet to be explored.

## Figures and Tables

**Figure 1 ijms-22-07467-f001:**
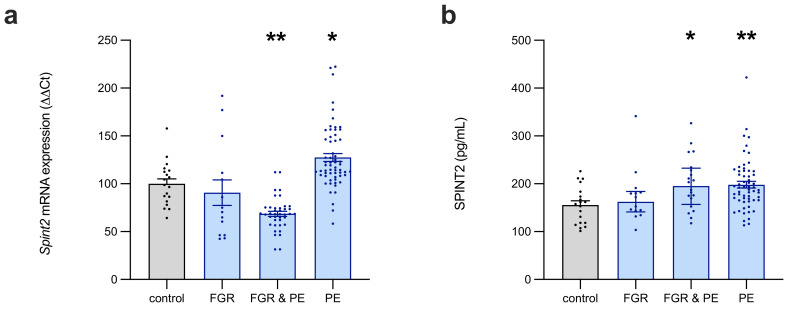
SPINT2 expression in placentas of patients with established placental disease. Compared to preterm controls, SPINT2 mRNA expression (**a**) was not altered in placentas from normotensive pregnancies affected by foetal growth restriction (FGR), but it was significantly decreased in those from pregnancies compromised by concurrent preeclampsia (PE) and FGR, and increased in preeclamptic placentas of AGA infants. SPINT2 protein expression (**b**) in these same placentas was also not changed in FGR-affected normotensive pregnancies, but was significantly elevated in all PE cases (with and without FGR). Each data point represents an individual patient sample; data are expressed as median ± IQR; * *p* < 0.05, ** *p* < 0.01.

**Figure 2 ijms-22-07467-f002:**
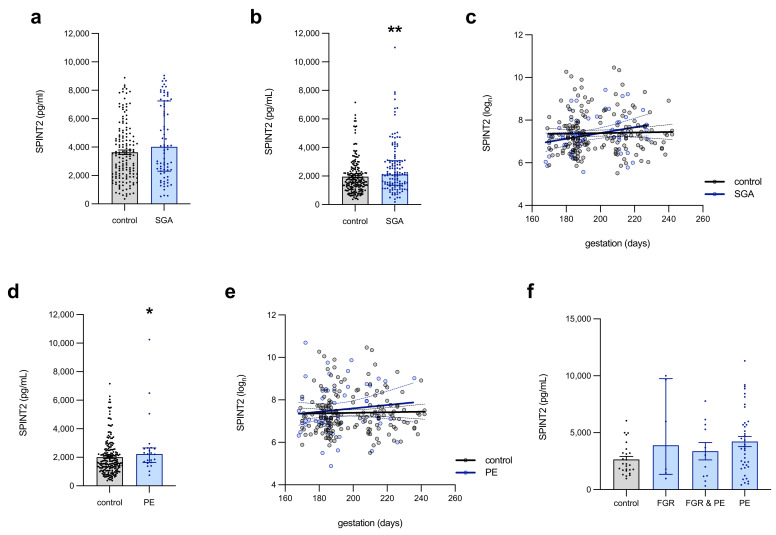
Circulating SPINT2 levels preceding preeclampsia development or birth of a small-for-gestational-age infant. In the blood of women on the day of delivery (**a**), SPINT2 protein expression was increased (approaching statistical significance, *p* = 0.051) in those who delivered an SGA infant, compared to AGA controls. This association was stronger at 36 weeks’ gestation, in which there was a significant elevation of SPINT2 levels among women who later delivered a small-for-gestational-age (SGA) infant (**b**) as well as in those women who subsequently developed PE (**d**), although the significance of the latter was lost when accounting for outliers. Earlier in the pregnancy, however, at 24–34 weeks, there was no association between circulating SPINT2 and SGA (**c**) nor PE (**e**) cases in samples from women with underlying vascular disease. In this cohort, SPINT2 did not fluctuate across gestation in controls nor PE; however, there was an apparent increase in SPINT2 across gestation in those women destined to birth an SGA infant. In the plasma collected on the day of delivery from women with diagnosed placental insufficiency (**f**), circulating SPINT2 was unchanged in cases, relative to controls. Each data point represents an individual patient sample; data are expressed as median ± IQR; linear regression showing 95% confidence intervals; * *p* < 0.05, ** *p* < 0.01.

**Figure 3 ijms-22-07467-f003:**
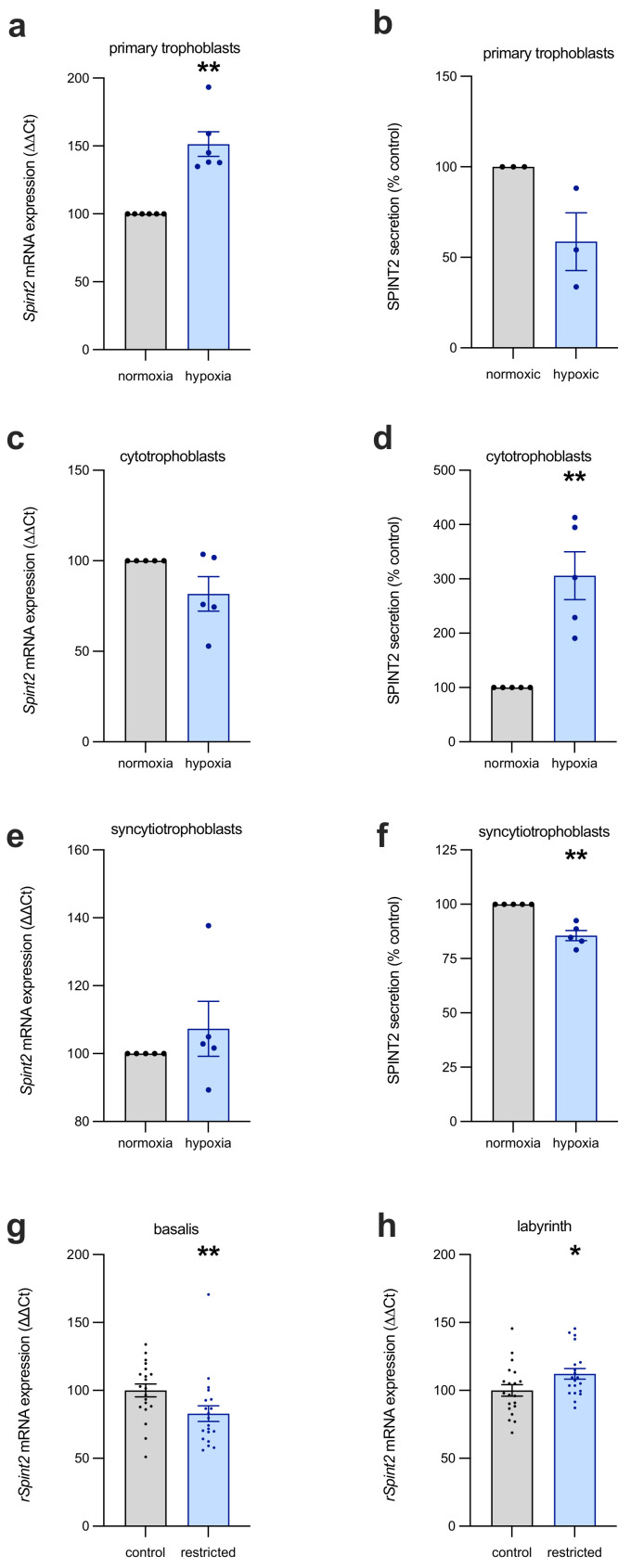
The effect of hypoxia on SPINT2 expression in placental cells. SPINT2 mRNA and protein secretion was measured in the following three types of trophoblast cultures: primary trophoblasts isolated from term placentas, and first-trimester cytotrophoblast and syncytiotrophoblast from a stem cell line. In the term primary trophoblasts (**a**), SPINT2 transcripts were significantly increased in response to hypoxia, while secreted protein levels (**b**) were not changed. Hypoxic conditions caused no alteration to the first-trimester cytotrophoblast stem cell SPINT2 mRNA (**c**), but did significantly increase the levels of SPINT2 secretion (**d**) compared to normoxic controls. The syncytialised first-trimester stem cells had no change in mRNA (**e**), although they did demonstrate decreased SPINT2 secretion (**f**). Experiments were repeated *n* = 3–5 times; data are expressed as mean ± SEM. In placentas from a rat model of uteroplacental insufficiency, changes were identified in rat SPINT2 mRNA (*rSpint2*) expression in both the basalis (**g**) and labyrinth (**h**) zones, being depressed and elevated, respectively. Each data point represents an individual rat placenta; data are expressed as median ± IQR; * *p* < 0.05, ** *p* < 0.01.

**Figure 4 ijms-22-07467-f004:**
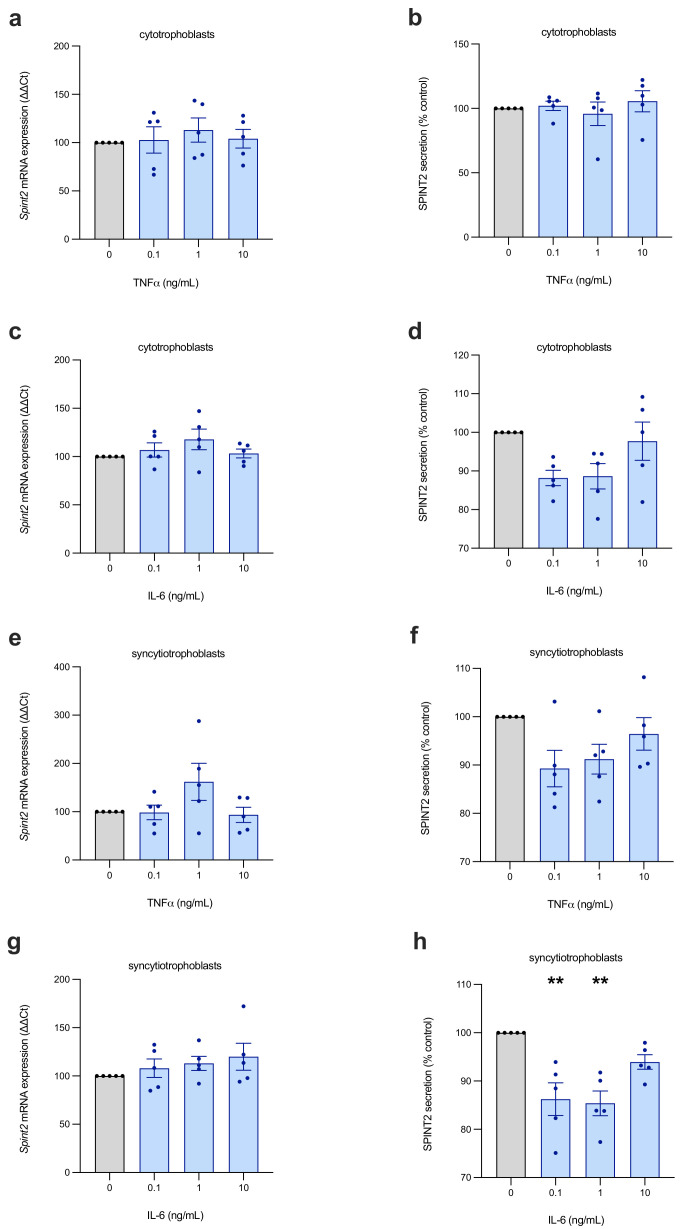
Inflammatory cytokine regulation of SPINT2 in placental cells. First-trimester cytotrophoblasts (**a**–**d**) and syncytiotrophoblasts (**e**–**h**) were treated with inflammatory cytokines, TNF or IL-6, at various doses. No significant changes were observed in SPINT2 mRNA (**a**,**c**,**e**,**g**) expression, nor in secreted SPINT2 (**b**,**d**,**f**); with the exception of IL-6–treated syncytiotrophoblasts (**h**), which stimulated a modest, but significant, decrease in SPINT2 secretion at lower doses. Experiments were repeated *n* = 5 times; data are expressed as mean ± SEM; ** *p* < 0.01.

## Data Availability

Raw data are available upon reasonable request from the corresponding author.

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
