# Peer review of "Elevated Circulating and Placental SPINT2 Is Associated with Placental Dysfunction"

_ijms, 2021, doi:10.3390/ijms22147467_

Round 1

Reviewer 1 Report

The manuscript submitted by Murphy and colleagues aims to determine the expression of SPINT2 during FGR and PE. The authors recently identified SPINT1 as a biomarker of placental dysfunction and hypothesize that its related protein SPINT2 may exhibit a similar role. Overall, the study is well done, but some minor issues need to be addressed.

  1. Most of the figures show the data as a dot plot of individual samples, however, figure 3a-f and figure 4 eschew this pattern and eliminate the dots. If possible, please include the dots to maintain consistency and transparency.
  2. Figures 3 and 4 are quite difficult to follow since the graphs have similar axis titles. Please include labels to differentiate the various samples tested in each panel. This is particularly important in figure 3 where the authors switch from human samples (panels a-f) to rat samples (panels g-h), yet the y-axes are identical. Rat genes should be denoted differently than human genes. This should also be addressed in the text, where human and rat genes are denoted the same.
  3. In figure 1 the authors show that SPINT2 mRNA is decreased in the FGR + PE cohort, but protein is elevated. On the other hand, in the PE cohort, both mRNA and protein are elevated. This point is only briefly touched upon in the discussion, but really warrants further exploration. Many transcript-protein pairs behave discordantly due to quirks of processing or trafficking, but it is more unusual for the same transcript-protein pair to be discordant in one condition (FGR + PE), but concordant in another condition (PE). The authors could further expand on this phenomenon in the discussion. Does this occur for other transcript-protein pairs in these cohorts?
  4. Tables 1 and 2 present multivariate regression models of logSPINT2 values accounting for gestational age and hypertensive status. It is not clear to me why these analyses are included, particularly since figure 2 establishes no relationship between gestational age and logSPINT2 levels. The multivariate models similarly show a null relationship and do not add much beyond the initial simple linear regressions in figure 2. Additionally, these analyses are not mentioned in the methods section.
  5. Figures 2b and 2d appear to have outliers which may be influencing the statistical significance achieved. The authors should conduct an outliers test (such as Grubbs test) to properly account for these extreme data points.
  6. It may be helpful to include a diagram of cytotrophoblast vs. synctiotrophoblast and rat basalis vs. labyrinth zones. This would help readers appreciate why these different sources may exhibit different responses.

Reviewer 2 Report

To authors,

The theme is worthy and the paper is well written. I have some minor (small) advices.

  1. In Introduction, you state some data regarding “cytotrophoblastic” SPINT1. I believe that this statement, and also the present study-context (for SPINT2), targeted to extravillous cytotrophoblast and not villous trophoblast. As a placentologist, this is evident but not every reader are placentologists, please describe this context. I mean that there are two types of cytotrophoblasts; villous and extra-villous. Please shortly add this statement.
  2. Abbreviations: PE/FGR; you defined them earlier but throughout the manuscript preeclampsia and fetal growth restriction (without abbreviations) appear. Please be consistent.

The authors faithfully confessed that SPINT2 is unlikely to be a clinically useful biomarker but stated “there is merit in further exploring the role of SPINT2 in placental development”. This may be a self-evident statement. Every substance, only if they have some associations (theoretically, clinically, empirically, etc.) with placental functions, may have some such associations (here, you state “development” or “placental function”). The concluding remarks should be more straightforward and practical. The final remarks might be made (with a little exaggeration) irrespective of the present study data.   
